# Biodereplication of Antiplasmodial Extracts: Application of the Amazonian Medicinal Plant *Piper coruscans* Kunth

**DOI:** 10.3390/molecules27217638

**Published:** 2022-11-07

**Authors:** Pedro G. Vásquez-Ocmín, Jean-François Gallard, Anne-Cécile Van Baelen, Karine Leblanc, Sandrine Cojean, Elisabeth Mouray, Philippe Grellier, Carlos A. Amasifuén Guerra, Mehdi A. Beniddir, Laurent Evanno, Bruno Figadère, Alexandre Maciuk

**Affiliations:** 1Université Paris-Saclay, CNRS, BioCIS, 91400 Orsay, France; 2Institut de Chimie des Substances Naturelles CNRS UPR 2301, Université Paris-Saclay, 1 Avenue de la Terrasse, 91198 Gif-sur-Yvette, France; 3Département Médicaments et Technologies pour la Santé (DMTS), CEA, SIMoS, Université Paris-Saclay, F-91191 Gif-sur-Yvette, France; 4Unité Molécules de Communication et Adaptation des Microorganismes (MCAM, UMR 7245), Muséum National d’Histoire Naturelle, CNRS, Sorbonne Universités, CP52, 57 Rue Cuvier, 75005 Paris, France; 5Dirección de Recursos Genéticos y Biotecnología (DRGB), Instituto Nacional de Innovación Agraria (INIA), Avenida La Molina N° 1981, La Molina, Lima 15024, Peru

**Keywords:** *Piper coruscans* Kunth (Piperaceae), biodereplication, heme binding, mass spectrometry, *Plasmodium*

## Abstract

Improved methodological tools to hasten antimalarial drug discovery remain of interest, especially when considering natural products as a source of drug candidates. We propose a biodereplication method combining the classical dereplication approach with the early detection of potential antiplasmodial compounds in crude extracts. Heme binding is used as a surrogate of the antiplasmodial activity and is monitored by mass spectrometry in a biomimetic assay. Molecular networking and automated annotation of targeted mass through data mining were followed by mass-guided compound isolation by taking advantage of the versatility and finely tunable selectivity offered by centrifugal partition chromatography. This biodereplication workflow was applied to an ethanolic extract of the Amazonian medicinal plant *Piper coruscans* Kunth (Piperaceae) showing an IC_50_ of 1.36 µg/mL on the 3D7 *Plasmodium falciparum* strain. It resulted in the isolation of twelve compounds designated as potential antiplasmodial compounds by the biodereplication workflow. Two chalcones, aurentiacin (1) and cardamonin (3), with IC_50_ values of 2.25 and 5.5 µM, respectively, can be considered to bear the antiplasmodial activity of the extract, with the latter not relying on a heme-binding mechanism. This biodereplication method constitutes a rapid, efficient, and robust technique to identify potential antimalarial compounds in complex extracts such as plant extracts.

## 1. Introduction

The ever-growing resistance of *Plasmodium* spp. to antimalarials continues to make it a global health problem [1,2]. A major factor that severely hinders the efforts to ‘roll back malaria’ is the emergence and spread of parasites resistant to affordable antimalarial agents [3]. Research of novel antimalarial molecules usually involves the screening of compounds obtained by medicinal or combinatorial chemistry or isolated from natural sources, followed by improvement of these molecules by hemisynthesis to enhance the biological activity profile. Finding a hit in a natural extract remains a difficult task; however, owing to the considerable progress made in compound isolation and identification techniques during the past decades, a combination of chemical and computational tools currently allows one to efficiently “dereplicate” an extract to identify the known molecules [4,5]. One stunningly performant example of dereplication is molecular networking (MN), a computer-based approach allowing one to visualize and organize tandem mass spectrometry datasets and automate a database search for specialized metabolites identified within complex mixtures [6,7,8]. Dereplication approaches strictu senso do not unequivocally identify compounds but rather annotate them as being identical to a known compound with a high probability. Such approaches also do not convey any biological information. We propose the term “biodereplication” as the early detection of a complex mixture of compounds responsible for the activity before any isolation or formal identification. Biodereplication approaches encompass TLC bioautography [9], LC online assays, and affinity-based methods [10] and require the choice of a relevant biological target to be implemented.

Heme (ferriprotoporphyrin IX, or Fe(III)PPIX) is freed upon the digestion of hemoglobin by the parasite. It is a strong oxidizing agent released inside the *Plasmodium* digestive vacuole (PDV) after hemoglobin proteolysis [11] and inactivated by the parasite by its sequestration into hemozoin crystals. The mechanism of action of quinoline drugs such as chloroquine is based on the inhibition of the crystallization pathway of free heme. The PDV is a multiphasic mixture of neutral lipids in the form of nanospheres suspended in an aqueous solution of proteins, at a pH ranging from 4.8 to 5.5 [12,13,14]. It is an acidic proteolytic compartment crucial for the metabolism of the parasite. In this vacuole, amino acids are released from globin, oxygen radicals are detoxified, drugs are accumulated, and free iron is generated [15]. Due to the important roles of heme in the elementary cellular processes of most organisms and especially in *Plasmodium*, heme metabolism has historically been a major target for antiparasitic drugs and is still considered a promising target for new molecules [16,17,18]. Understanding the heme crystallization process has been a pivotal point in characterizing the mechanism(s) of action of the major quinoline antimalarial drugs [19] and is crucial for the development of new drugs able to overcome parasite resistance [20]. The formation of π–π non-covalent bonds is a necessary requirement for the inhibition of heme crystallization by antimalarial drugs [21,22]. Various chemical and physical factors modulate hemozoin formation, such as the hydrophobicity of native, surrounding alcohols and lipids in the digestive vacuole as they impact the solubilization of heme aggregates or reduce surface tension.

So far, several methodologies have been used for the detection of compounds targeting the heme detoxification pathway for drug discovery purposes, i.e., spectrophotometric microassay of heme polymerization (HPIA) [22,23], nuclear magnetic resonance (NMR) [24], or the detection of heme adducts via mass spectroscopy (MS) [25,26]. Indeed, ESI provides a rapid, sensitive, and highly selective tool for probing non-covalent interactions [27,28]. These methods are based on the assumption that a molecule binding to heme would prevent its crystallization and increase the amount of free heme in the digestive vacuole, hence leading to parasite death. Stable non-covalent complexes can be formed between Fe(III)-heme and antimalarial agents, i.e., quinine and artemisinin and its derivatives dihydroartemisinin, α- and β-artemether, and arteether [11]. Relative binding strengths between drugs and Fe (III)-heme can also be measured. This approach has previously been shown to be useful in determining the structure–activity relationship of antimalarial agents, namely, terpene isonitriles and neocryptolepine derivatives [29], or in assessing the strength of π–π non-covalent binding [25,26,30].

In this work, we propose an integrated workflow applicable to natural extracts and dedicated to isolating compounds of biological relevance. We implement several tools to screen an extract for heme-binding compounds and annotate and isolate them in an integrated workflow. This allows the early detection of potential antiplasmodial compounds in crude extracts. Heme binding used as a surrogate of antiplasmodial activity is monitored by mass spectrometry under biomimetic conditions. Automated annotation of targeted mass through molecular networking and data mining was followed by mass-guided compound isolation by taking advantage of the versatility and finely tunable selectivity offered by centrifugal partition chromatography. This biodereplication workflow was applied to an extract of *Piper coruscans* Kunth (Piperaceae), a plant traditionally used in the Peruvian Amazonia as a medicinal plant to fight malaria and whose aerial part ethanolic extract showed promising in vitro activity on *P. falciparum* (IC_50_ = 1.36 µg/mL) [31,32].

## 2. Results and Discussion

### 2.1. Biodereplication as a New Strategy of Antiplasmodial Drug Discovery

In this study, we describe a biomimetic, miniaturized, automated, MS-based heme-binding assay and data-mining workflow, which improves on the work described by Muñoz-Durango et al. [25]. It reduces false positives and allows for a high-throughput screening format. The overall workflow is presented in Figure 1. Compared to the previous version of the assay, based on aqueous basic conditions, it benefits from a biomimetic approach by mimicking the pH (pH 5.2) and multiphasic nature (aqueous–lipid interface) of the PDV [33]. Heme is insoluble in acidic water as the protonation of its propionic residues decreases aqueous solubility and increases intermolecular hydrogen bonding [34]. The value of pH may also indirectly affect the formation of π–π complexes [22]. Consequently, the lower the pH, the lower the solubility but the higher the proportion of dimers. Additionally, recent reports suggest that hemozoin formation may occur at the interface of aqueous and lipidic regions in the PDV [34,35,36]. Data from X-ray tomography of the digestive vacuole indicate that, while lipid nanospheres are important for hemozoin nucleation, crystal growth may occur in the aqueous phase [37,38]. Regarding the PDV biological lipid mixture being solid at room temperature (it is liquid at 37 °C), *N*-octanol has been suggested as a more convenient biomimetic model than the glycerides mixture [39,40,41]. The solubility of water is lower in *N*-octanol than in a glycerides mixture, but it is still sufficient for hydrogen bond formation and hematin crystal formation [35]. The organic phase of the citric-buffer-saturated octanol system (CBSO) was found to properly solubilize heme [33,42]. Octanol mimics the lipid nanospheres, while the solubilized citric buffer imitates the acidic environment of the PDV [13,33]. After having screened different conditions (pH 4.8, 5, 5.2, and 10 ad substitution of citric acid with formic acid or ammonia) and comparing the effects on hemin solubilization and the intensity of adducts signals, we used a modified CBSO solution of pH 5.2, allowing the intensities of the adduct signals to be equal or superior to those observed in aqueous basic conditions.

Due to the nature of the solution composed of octanol, DMSO, and water, parameters of the ionization source were optimized to allow evaporation, ionization, and low background noise (temperature and gas flow of 325 °C and 11 L/min, respectively). To ease the formation of the micellar interface, we added a small concentration of Tween 20 detergent [43]. The MS analyses of such a mixture of heme with a complex extract show numerous peaks (Figure 2). Deciphering such signals to detect adducts can be performed by several methods. One can compare the spectrum from the extract alone with the spectrum of the mixture of the extract plus heme (*m*/*z* 616) and visually detect whether new ions of *m*/*z* > 616 appear in the second case. This approach is definitely time-consuming and prone to mistakes and failures. As an alternative, a program script could be designed to analyze data and automatically detect such discrepancies. Another method consists of using a triple-quadrupole instrument in precursor ion mode. In such an implementation, with the second quadrupole set to detect *m*/*z* 616, any adduct analyzed by the first quadrupole and able to produce heme as a fragment upon dissociation would be highlighted. Unfortunately, such instruments do not provide high *m*/*z* accuracy and are not able to give any information on the molecular formula of unknown compounds. In contrast, a Q-ToF instrument provides high-accuracy readings but is not able to work in precursor ion mode. Indeed, it needs to gather ions in a “pulse” before flight-time measurement and thus cancels the “separation in time” of precursor ions by the two first quadrupoles. A Q-ToF instrument can indicate when a target fragment ion is produced but is unable to indicate to which precursor it correlates. This limitation can be artificially minimized when running chromatographic separation, which is not applicable in our case since the injection is performed in infusion mode (see below). To maintain the advantage of QToF’s high resolution and still be able to deconvoluate the adduct data into compound information, we applied a molecular networking (MN) approach. In such an approach, a network is computerized by the GNPS platform [8] or equivalent software to detect ions producing heme fragments upon dissociation. In such an analysis, parameters can be set up so every species producing *m*/*z* 616 as a fragment is plotted as a node in the same network cluster. Nodes in a cluster are connected following their cosine values. The cosine value approaches 1 when fragmentation routes are very similar [7].

Two samples were used as a validation procedure. An equimolar mix of eight structurally diverse antimalarial drugs known to bind to heme was used as a positive control to validate the ability of the workflow to detect heme ligands. An alkaloid extract of Cinchona pubescens Vahl (Rubiaceae) containing quinine (a heme ligand) along with other heme-binding and non-heme-binding alkaloids was used to validate the ability of the workflow to discriminate binding and non-binding compounds in a complex extract. In all cases, along with heme and heme dimers, heme fragments were identified at *m*/*z* 557 and 498 corresponding to the loss of the first and second carboxymethyl groups, [heme *−* CH_2_COOH]^+^ and [heme *−* (CH_2_COOH)*_2_*]^+^ ions, respectively [44]. All the antimalarial drugs showed adducts (Figure 2A) with heme-Fe(III). After MN data processing, every drug of the mix of antimalarial drugs was detected in the same cluster, although these drugs do not belong to the same chemical family. The heme adduct cluster from the alkaloid extract (Figure 2B) showed all compounds known to bind to heme (see Appendix A), with some of the adducts (e.g., *m/z* 956) possibly resulting from the binding of heme with quinoline artifacts [45,46]. Noteworthily, a non-covalent bis-adduct was observed for quinine [(2heme-Fe(III) – H) + quinine]^+^ at *m/z* 1556.539 (see Appendix A). Docking studies on hemozoin-quinoline drugs also reported such a sandwich adduct with quinine and chloroquine [47]. Nevertheless, in our experiment, this adduct was not deconvoluted by molecular networking into a heme-binding compound. This may be explained by the fact that these adducts are dissociated at very high collision energies. Lastly, under acidic and reducing conditions (obtained by the adjunction of glutathione), several artemisinin-heme-Fe(II) adducts were detected at *m*/*z* 838, 898, and 1180 (see Appendix A). The adduct at *m/z* 838 [heme + artemisinin -CH_2_-COOH + H]^+^ was shown to be of a covalent nature by MS–MS analyses, as reported in the literature [24]. The reductive activation of endoperoxide by Fe(II) heme produces homolytic cleavage of the endoperoxide bond and the subsequent formation of the artemisinin-heme adduct is able to alkylate heme or other proteins [25].

### 2.2. Scientific Validation of the Traditional Use of P. coruscans as an Antimalarial Remedy Using Biodereplication

The biodereplication workflow was applied to an ethanolic extract of leaves of *P. coruscans* (PCE), an Amazonian plant traditionally used against malaria [31]. MS analysis of the PCE–heme mixture showed four *m/z* of interest (Figure 3A, left panel): 914.2938, 900.2814, 886.5327, and 871.5579. As expected, heme dimers were also identified in the same spectrum (Figure 3A, right panel). In the molecular network (Figure 3B), nodes within the adduct cluster are related to satisfactory cosine values ranging from 0.7 to 0.96. In parallel, LC–MS data of PCE were analyzed to annotate the targeted *m*/*z* using databases (Figure 3C). This analysis confirmed that the adduct at *m*/*z* 871.55 (C_16_H_17_NO_2_) corresponds to a minor compound at *m*/*z* 256.13 [M+H]^+^ (Figure 3D1,D2). The adducts observed at *m*/*z* 914.28 (C_18_H_18_O_4_), 900.28 (C_17_H_16_O_4_), and 886.53 (C_16_H_14_O_4_) correspond to compounds in the extract at nominal *m*/*z* 299, 285, and 271, respectively, which appeared to encompass several isobaric species present in the extract (Figure 3D3–D5). All the annotated compounds and their details are shown in Table 1. Although isomers are distinguished on the LC–MS data (Figure 3D1–D5), this is not possible in the heme-binding assay, which is performed in infusion mode. Applying a chromatographic separation to the heme–extract mixture would pose additional problems: Among others, it may displace the equilibrium of the non-covalent association or lead to an absence of separation of the different adducts. Injection after rapid mixing and short incubation guarantees the persistence of non-covalent bounds even of low strength. Furthermore, the retention time of adducts with different isomers would likely not correlate with the retention times of the isomers alone, so the issue of attributing *m*/*z* to a given eluted peak would be displaced and not solved. Hence, we did not address these questions, as our aim is rapid, high-throughput format visualization of non-covalent binding without chromatographic separation. The molecular network based on LC–MS of the ethyl acetate fraction (PCA) (see Appendix A) allowed us to determine three main clusters of the phytochemical superclass: Compounds 2, 3, 6, 7, 8, and 9 (flavonoids), 11 (alkaloid derivative), and 12 (naphthalene derivative).

Target compounds identified by the MS-binding assay from *P. coruscans* (Figure 3E) were displayed at *m*/*z* 299 (compounds 1, 5, and 9), *m*/*z* 285 (compounds 4, 6, 2, 10, and 8), *m*/*z* 271 (compounds 3 ad 7), and *m*/*z* 256 (compound 11). Annotation resulted in the identification of the chalcones aurentiacin (1), stercurensin (2), and cardamonin (3); the flavanones strobopinin 7-methyl ether (4), 5-hydroxy-7-methoxy-6,8-dimethyl flavanone (5), desmethoxymatteucinol (6), alpinetin (7), pinocembrin (8), and dimethyl cryptostrobin (9); and an alkylamide, *N*-benzoyltyrarnine methyl ether (11). Among the ten targeted compounds annotated, three were annotated at the “genus” annotation level (i.e., picked in the database of compounds described in the literature as identified in the *Piper* genus). Cardamonin (3), 5-hydroxy-7-methoxy-6,8-dimethyl flavanone (5), and alpinetin (7) were previously isolated from *Piper aduncum*, *P. hispidum*, and *P. hostmannianum*, respectively [50,51]. Compound 10 was not annotated because it is described as resulting from degradation in the MS source. A suggested MS source decomposition pathway for this compound is proposed in Appendix A. Species from the Piperaceae family (e.g., *Piper* and *Peperomia*) are aromatic plants used in traditional medicine in tropical and subtropical regions [52]. Alkaloids or amides such as piperine (being neuroprotective) [53], piplartine (showing anticancer properties and antiparasitic against *Schistosoma mansoni*) [54,55], aduncamide (being antibacterial against *Bacillus subtilis* and *Micrococcus luteus*) [56], or benzoic acid derivatives (being antiplasmodial against *P. falciparum*) [57] are only a few examples of the active compounds that validate these uses of *Piper* species. Compounds isolated from the *Piper* genus may also include propenylphenols, lignans, neolignans, terpenes, steroids, kawapyrones, piperolides, flavones, chalcones, and dihydrochalcones [58].

Once *m*/*z* values of target compounds were annotated (Table 1), we performed mass-guided isolation. Target species were detected principally in the ethyl acetate fraction, except for the non-polar compounds (1, 4, and 5), which were detected in the cyclohexane fraction. We isolated these molecules by CPC using a mixture of five solvents specifically optimized for this purpose. The choice of a CPC solvent system is guided by the solubility of the analytes, as well as their partition coefficient. In the preliminary optimization step, the rational selection of suitable stationary and mobile phases is essential [59]. This particular solvent mixture (heptane/ethyl acetate/butanol/methanol/water) was selected as it provided a good retention factor and selectivity for the targeted molecules, namely, a partition coefficient of nearly 1 for the targeted analytes and different from 1 for other analytes. Two solvent systems were designed, the first system suited to non-polar compounds with heptane/ethyl acetate/butanol/methanol/water (12:14:14:23:37) and another system suited to moderately polar compounds with heptane/ethyl acetate/butanol/methanol/water (8:16:16:18:42) with increased proportions of ethyl acetate and butanol in the organic phase. Non-polar compounds 1, 4, and 5, were isolated by CPC, as can be seen in the CPC fractograms (See Appendix A). For polar compounds, LC–MS analysis of both phases of several biphasic systems allowed us to determine that the solvent system heptane/ethyl acetate/butanol/methanol/water (20:10:10:30:30) showed satisfactory selectivity targeted on flavonoid compounds such as flavanones or chalcones. This system is a variant of the Oka range in which hexane was replaced by heptane [59]. The use of this system allowed us to obtain compounds 1, 4, and 5 in only one step (injection of the fraction into the CPC system and the collection of pure compounds). Other compounds required a final step by preparative HPLC. More information on the identification of the compounds is given in Appendix A, i.e., 1D and 2D NMR (see Appendix A).

The antiplasmodial activity of PCE was promising (IC_50_ = 1.36 µg/mL) [31]. The chalcones aurentiacin (1) and cardamonin (3) showed the best antiplasmodial activity with IC_50_ = 2.25 and 5.5 µM, respectively. Aurentiacin has the lowest cytotoxicity on fibroblasts and the highest selectivity index, suggesting it is the most interesting structure (Table 2). Considering the low DV_50_ value for these two compounds, namely, 5.15 and 5.60 eV, respectively, their antiplasmodial activity is likely not related to a heme-binding mechanism, corroborating hypotheses from other authors [60]. Indeed, DV_50_ is the fragmentor voltage at which half of the adduct is dissociated (see Figure 4). We previously demonstrated that it can be interpreted as proportional to the binding strength between the heme and the compound [25,26]. As the isolation of compounds was mass-driven, isomers could not be distinguished, and compounds not binding to heme were also isolated. Antiplasmodial activity may still be detected for these compounds, albeit not mediated by heme binding. Cardamonin was reported to have significant activity against *Trypanosoma brucei* (IC_50_ = 1.8 µM), a parasite that does not rely on hemoglobin for its metabolism [61]. Studies also showed that cardamonin binds with a high affinity to site II of HSA (human serum albumin) [62]. Chalcones are abundant in edible plants and are precursors of flavonoids and isoflavonoids. The presence of a double bond conjugated with the carbonyl is believed to be responsible for the biological activities of chalcones as the removal of the double bound has an inactivating effect [63]. The antimalarial activity of chalcones has triggered great interest, and several natural and synthetic chalcones have been described to possess antimalarial effects [64]. Their antimalarial activity is supposed to be a result of a Michael addition of cellular nucleophilic sites to the activated double bond [65]. Indeed, the Michael acceptor site of chalcones can readily form covalent bonds with the sulfhydryl of cysteines or other thiols to obtain a Michael adduct, which may play an important role in the biological activities of chalcones [66]. Covalent heme-chalcone adducts via the Michael addition pathway have not been reported so far. Nevertheless, the asymmetric Michael addition of nitroalkanes to chalcones has been reported [64]. Licochalcone A is an example of a chalcone inhibiting the in vitro growth of both chloroquine-sensitive (3D7) and chloroquine-resistant (Dd2) strains of *P. falciparum*. It was shown to have multiple targets in the *P. falciparum* mitochondrion. It inhibited ubiquinol cytochrome c reductase (bc1 complex) and succinate ubiquinone reductase (complex II) [60]. It is known that chalcones bearing a Cl group in position 4 of ring A and methoxy (position-3) and allyloxy (position-4) on ring B are active against 3D7 *P. falciparum* with average IC_50_ values of 2.5 µM [67,68].

The *N*-benzoyltyrarnine methyl ether (11) compound has the highest DV_50_ value (Figure 4) but, surprisingly, showed no antiplasmodial activity, suggesting that DV_50_ values should only be considered as relevant when they are above the DV_50_ value of reference drugs, e.g., chloroquine. All the isolated flavanones (4–9) showed moderate antiplasmodial activity, ranging from 85 to 33 µM (Table 1). This activity can be partly explained by their cytotoxicity, as their selectivity indexes are very low. Their DV_50_ values were in accordance with this moderate activity, as these compounds would not bind or only bind loosely (DV_50_ in the range of 8–9) while chloroquine had a DV_50_ above 14. The DV_50_ value of the artemisinin covalent adduct is similar to that of chloroquine but the profile of the stability curve is not sigmoidal (Figure 4), with this observation being explained by the fact that the observed effect is not a dissociation but an actual fragmentation with covalent-bound breakage. Adducts at *m*/*z* 898 [heme + artemisinin]^+^ and 1180 [heme + 2 artemisinin]^+^ were identified for the first time by MS as being non-covalent adducts as they would dissociate very quickly and have a DV_50_ < 2 (data not shown).

## 3. Materials and Methods

### 3.1. Biological Material

Leaves of *Piper coruscans* Kunth (*Piperaceae*) were collected by Dr. Pedro Vásquez-Ocmín in December 2016 in Lupuna (Zona II) village, in the province of Maynas, located in the region of Loreto, Peru (18M 0704173, UTM 9585656). The plant was identified by Dr. Carlos Amasifuén and deposited in the Herbarium Amazonense of the National University of the Peruvian Amazon (AMAZ), Iquitos, Peru, with the voucher number 039849. Leaves were dried and ground (600 g), soaked for 24 h twice in 5 L of ethanol 96%, and filtrated, and the extract was evaporated under reduced pressure at below 40 °C.

### 3.2. Heme-Binding Assay by Mass Spectrometry (MS)

Hemin (iron-containing porphyrin with chlorine counterion) and the crude ethanolic extract were dissolved in DMSO at 5 mM and 10 mg/mL, respectively. Pure compounds were dissolved in DMSO at the concentration of 10 mM, except for Tween 20, which was dissolved in water at 10 µM. Citric-buffer-saturated octanol (CBSO) [70] was prepared by mixing 5 mL of citric acid, 50 mM and pH 5.2, and 15 mL of *N*-octanol (anhydrous, ≥99%) and letting it settle at 23 °C for 30 min. The upper phase of the CBSO biphasic system was used. Autosampler 1260 infinity G1367E 1260 Hip ALS (Agilent Technologies, Santa Clara, CA, USA) was used to perform the automatic mixing of aliquots in 384-well plates. The incubation mixture was as follows: Compound 5 µL + heme 5 µL + Tween 5 µL + organic phase of CBSO 85 µL. Samples were injected in infusion mode (direct injection) in a 6530 Accurate-Mass QToF LC–MS instrument (Agilent Technologies) with the following settings: Positive ESI mode and 2 GHz acquisition rate. Ionization source conditions were a drying gas temperature of 325 °C, a drying gas flow rate of 11 L/min, a nebulizer of 35 psig, a fragmentor of 175 V, and a skimmer of 65 V. The range of *m*/*z* was 200–1700. In positive-ion mode, purine ion C_5_H_4_N_4_ [M + H]^+^ (*m*/*z* 121.050873) and the hexakis (1H,1H,3H-tetrafluoropropoxy)-phosphazene ion C_18_H_18_F_24_N_3_O_6_P_3_ [M+H]^+^ (*m*/*z* 922.009798) were used as internal lock masses. Full scans were acquired at a resolution of ca 11 000 (at *m*/*z* 922).

On the spectral window, heme appeared at *m*/*z* 616.16; adducts at *m*/*z* 616.16 + “X” and 2heme + “X”; heme + DMSO at *m*/*z* 694.19; heme dimers at *m*/*z* 1231.32 [2heme − H]^+^; *m*/*z* 1253.30 [2heme − 2H + Na]^+^; *m*/*z* 1267.26 [2heme + 35Cl]^+^ and 1269.24 [2heme + 37Cl]^+^. Chloroquine (chloroquine diphosphate, Sigma-Aldrich, St. Louis, MO, USA) and artemisinin (Sigma-Aldrich) of 5 mM in water were used as positive control. To obtain heme-Fe(II) from hemin (heme-Fe(III)), glutathione (50 mM in water) was mixed with hemin (5 mM in DMSO) in a ratio of 1:1 and left to incubate for 40 min. This solution was used as the heme solution, as described above.

### 3.3. Collision-Induced Dissociation

Dissociation in the collision chamber was caused by increasing collision energies from 0 to 56 eV. The MS–MS targeted mode, an isolation width of *m*/*z* 1.3, and the flow injection analysis mode (FIA) were applied. MS–MS parameters were the same as described above. The results were plotted, and a sigmoidal regression curve allowed us to calculate the energy necessary to dissociate 50% of adducts (DV_50_) using GraphPrism software version 5.00 (San Diego, CA, USA) or a Python script [25].

### 3.4. Visualization of Adduct Fragmentation via Molecular Networking (MN)

Ethanolic extract and hemin (heme-Fe(III)) were mixed and injected using infusion mode for the MS–MS experiment. Three collision energies were used: 30, 50, and 70 eV. The ten most intense ions (top 10) per cycle were selected. MS–MS acquisition parameters were as follows: *m*/*z* range of 100–1700, default charge of 1, a minimum intensity of 5000 counts, 3 spectra/s, and isolation width of 1.3 *m*/*z*. A list of compounds was generated by the “find by auto MSMS” algorithm of MassHunter software (Qualitative Analysis B.07.00, Agilent Technologies). The “extract average MS–MS spectrum” mode was used to summarize and average the results of the 3 collision energies. Exportation of generated spectra was performed in MGF format using the option “entire data file”. The information generated was uploaded onto the online platform GNPS (global natural product social network) [71]. Parameters used for the algorithm were as follows: Precursor ion mass tolerance = 1; fragment ion mass tolerance = 0.3; min pairs cos = 0.7; minimum matched fragment ions = 3; and minimum cluster size = 1. From the data, a network was calculated measuring correlations and was plotted using Cytoscape 3.5.1. The *m*/*z* ratio of each adduct [heme + “X”]^+^ was observed as an individual node in the same cluster, related to other nodes in agreement with the loss of a similar fragment (heme at *m*/*z* 616.16). An alkaloid extract from *Cinchona pubescens* Vahl (*Rubiaceae*) and eight antiplasmodial drugs (chloroquine, quinine, amodiaquine, ketoconazole, miconazole, mefloquine, praziquantel, and sulfadoxine) were used as examples of the visualization of adduct fragments by MN. The methodology and conditions were the same as described above.

### 3.5. LC–MS of Extract and Compounds Annotation

The ethanolic extract from *P. coruscans* (PCE) at 5 mg/mL in DMSO was analyzed by liquid chromatography performed on an Agilent 1260 series HPLC coupled to a 6530 QToF (Agilent Technologies). Chromatography separations were performed on a Sunfire C18 column, 2.1 × 150 mm−3.5 μm (Waters). The mobile phase comprised water (0.1% formic acid) (A) and acetonitrile (B). A linear gradient of 5–100% B in 20 min at a constant flow rate of 0.2 mL/min was used to elute the extract, followed by 5 min of conditioning and 10 min of equilibration. Analyses of the samples (1 μL injected) were performed in an optimized DDA setting (280 nm) followed by MS–MS scans for the three most intense ions (Top 3). Mass spectrometer settings were as follows: Positive ESI mode, 50–3200 mass range calibration, and a 2 GHz acquisition rate. The ionization source conditions were as described above. MS–MS acquisitions were performed using three collision energies: 10, 20, and 30 eV. MS–MS acquisition parameters were defined as follows: *m*/*z* range of 100–1700, default charge of 1, a minimum intensity of 5000 counts, rate/time = 3 spectra/s, and isolation width of 1.3 u [32]. The resolution of the MS instrument was ca 5000 at *m*/*z* 118 and 10,000 at *m*/*z* 922. LC–MS data were processed with MS-DIAL version 4.80 [72]. MS1 and MS2 tolerances were set to 0.01 and 0.05 Da, respectively, in centroid mode for each dataset. Peaks were aligned with an RT tolerance of 0.2 min, a mass tolerance of 0.015 Da, and a minimum peak height detection at 1 × 10^4^. MS-DIAL data were deconvoluted with MS-CleanR [73] by selecting all filters with a minimum blank ratio set to 0.8 and a maximum relative standard deviation (RSD) set to 40 as previously reported [74,75]. Two peaks were kept in each cluster for further database requests and the retained features were annotated with MS-FINDER version 3.52 [76]. The MS1 and MS2 tolerances were set to 5 and 15 ppm, respectively. The formula finder was exclusively processed with C, H, O, and N atoms. Two pathways were followed for the compound’s annotation by comparing data to 2 dedicated compounds databases: (i) Compounds identified in the literature for both the *Piper* genus and the *Piperaceae* family (Dictionary of Natural Products version 28.2, CRC press) mined by MS Finder based on exact mass and in silico fragmentation (“genus” annotation level); and (ii) generic databases included in MS-FINDER such as PlantCyc, ChEBI, NANPDB, COCONUT, and KNApSAcK (“generic” annotation level).

### 3.6. Isolation of Targeted Compounds

Molecule isolation was performed using centrifugal partition chromatography (CPC) (Sanki LLB-M, 200 mL capacity), preparative HPLC, and LC–MS. Successive liquid–liquid extraction was carried out on 60 g of PCE using cyclohexane/water with a cyclohexane fraction (PCC) yielding 45 g, an ethyl acetate/water with ethyl acetate fraction (PCA) yielding 10 g, and water (PCAq) yielding 11 g. Targeted molecules were localized by LC–MS analysis in the cyclohexane and ethyl acetate extracts (as described in 2.5). CPC in descending mode was performed on 1.8 g of the PCC fraction using the biphasic solvent system heptane/ethyl acetate/butanol/methanol/water (20:10:10:30:30) determined by testing several biphasic systems and analyzing both phases by LC–MS (data not shown). Fifteen sub-fractions were obtained from PCC (PCC-1 to PCC-15). Fractions of 0.5 mL were collected, analyzed by LC–MS (as described in 4.6) to verify these fractions, and grouped into 15 sub-fractions. Isolated compounds were aurentiacin (1) (20 mg) from PCC-3; strobopinin 7-methyl ether (4) (18 mg); and 5-hydroxy-7-methoxy-6,8-dimethyl flavanone (5) (7 mg) from PCC-5 and PCC-8-10, respectively. Then, CPC in descending mode was performed on 8 g of the PCA fraction using the biphasic solvent system heptane/ethyl acetate/butanol/methanol/water (12:14:14:23:37) determined by testing several biphasic systems and analyzing both phases by LC–MS, with conditions similar to those described above. Six sub-fractions were obtained (PCA-1-to PCA-6), including PCA-2 (yielding 840 mg). A second CPC in descending mode was performed on 800 mg of PCA-2 with heptane/ethyl acetate/butanol/methanol/water (8:16:16:18:42), producing 13 sub-fractions (PCA-2-1 to PCA-2-13). Three preparative HPLC were performed on a Sunfire column (150 × 19 mm I.D., 5 µm (Waters) with eluent A (water + 0.1% formic acid) and eluent B (acetonitrile), from 30 to 70% B in 20 min from PCA-2-4 (26 mg), PCA-2-6 (65 mg), and PCA-2-13 (120 mg). Eight compounds were isolated: Desmethoxymatteucinol (6) (1.2 mg), 2′, 4′-dihydroxy-6′-methoxy-3′-methylchalcone (2) (0.8 mg), and cardamomin (3) (3.7 mg) from PCA-2-13; alpinetin (7) (1.1 mg) and compound (10) (1.3 mg) from PCA-2-4; and pinocembrin (8) (2.6 mg), dimethyl cryptostrobin (9) (0.6 mg), and *N*-benzoyltyramine methyl ether (11) (1.9 mg) from PCA-2-6. Along with these targeted compounds, one untargeted compound was isolated: 1H-inden-1-one (12) (3 mg) from PCA-2-4. Details of the compound identification by LC–MS (see Appendix A) and NMR spectroscopic data are given in Appendix A).

### 3.7. Antiplasmodial Activity

#### 3.7.1. *Plasmodium falciparum* Culture

The chloroquine-sensitive 3D7 *P. falciparum* strain (clone of the NF54) was obtained from the Malaria French National Reference Center (CNR Paludisme, Hôpital Bichat Claude Bernard, Paris). The strain was maintained in O+ human erythrocytes in an albumin-supplemented RPMI medium under continuous culture using the candle-jar method [77]. The parasites were synchronized to the ring stage via repeated sorbitol treatment [78].

#### 3.7.2. In Vitro Antiplasmodial Activity on *Plasmodium falciparum*

A 2.5% (*v*/*v*) erythrocytes suspension with 1% parasitemia (number of infected red blood cells per 100 red blood cells) was incubated in duplicate with the compounds at concentrations ranging from 48.5 nM to 100 µM, obtained by serial dilution. After 44 h of incubation at 37 °C, the plates were subjected to 3 freeze–thaw cycles to achieve complete hemolysis. The parasite lysis suspension was diluted 1:5 in lysis buffer. In vitro susceptibility is expressed as the concentration inhibiting 50% of the parasite’s growth (IC_50_). Parasite growth was determined using SYBR^®^ Green I, a dye with strong fluorescence enhancement upon contact with DNA. Incorporation of the SYBR^®^ Green I (Fisher Scientific, Illkirch-Graffenstaden, France) in parasite DNA was measured using the Master epRealplex cycler^®^ (Eppendorf, Montesson, France) according to the following program to increase the SYBR^®^ Green I incorporation: 90 °C (1 min) and a decrease to 10 °C over 5 min followed by fluorescence reading. Untreated infected and uninfected erythrocytes were used as controls and chloroquine diphosphate (Sigma, Lezennes, France) was used as a reference drug [31,32,79]. IC_50_ was calculated by IC-estimator online software (www.antimalarial-icestimator.net, accessed on 18 January 2018).

### 3.8. Cytotoxicity Studies

Cellular cytotoxicity was evaluated using the AB943 primary human dermal fibroblast cell line maintained at 37 °C in the RPMI 1640 medium supplemented with 10% (*v*/*v*) fetal calf serum. Cells were seeded in 96-well plates (20,000 cells/mL) and incubated for 24 h, then treated with the drug for 72 h. After incubation, the cell growth medium was replaced by 100 μL of RPMI 1640 containing 20% (*v*/*v*) Alamar Blue (Thermo-Fisher, Illkirch-Graffenstaden, France). Fluorescent viable cells were monitored after 5 h of incubation at 37 °C at a wavelength of 530 nm for excitation and 590 nm for emission in an FL600 luminescence spectrometer (Biotek, Colmar, France). CC_50_, corresponding to drug concentrations causing 50% AB943 cell proliferation inhibition, were calculated from the drug concentration–response curves [80].

### 3.9. ADMET Prediction

ChemSketch (ACD/Labs Products) software was used to predict the logP of all compounds, using the SK2 format as the file input format [81].

## 4. Conclusions

Crystallization of heme is the principal pathway used by *Plasmodium* to detoxify hemoglobin digestion by-products. Interruption of this process remains a promising target for antimalarial drug discovery as it does not lead to resistance per se. Our biodereplication methodology contributes to the automated detection of compounds binding to heme in complex mixtures without the need for maintaining a parasite culture. Its miniaturized format accounts for minimal reagent consumption and is amenable to the high-throughput format (up to 480 extracts analyzed per day). Applying a molecular networking approach combined with bio-informatic data-mining tools is an efficient strategy to analyze data and annotate and designate compounds binding non-covalently to heme. The approach was applied to a *Piper* extract, highlighting chalcones as the main compounds responsible for the antiplasmodial activity. The versatility and finely tunable selectivity of CPC was illustrated in the isolation process. This integrated biodereplication workflow able to detect and designate antiplasmodial compounds in an extract in 3 min represents a significant input to antimalarial drug discovery. We propose an integrated workflow involving the non-orthodox use of molecular networking to make such biodereplication readily implementable in every lab equipped with HRMS–MS spectrometry. The biodereplication method allows a rapid, efficient, and robust technique to identify potential antimalarial compounds in complex extracts such as plant extracts.

## Figures and Tables

**Figure 1 molecules-27-07638-f001:**
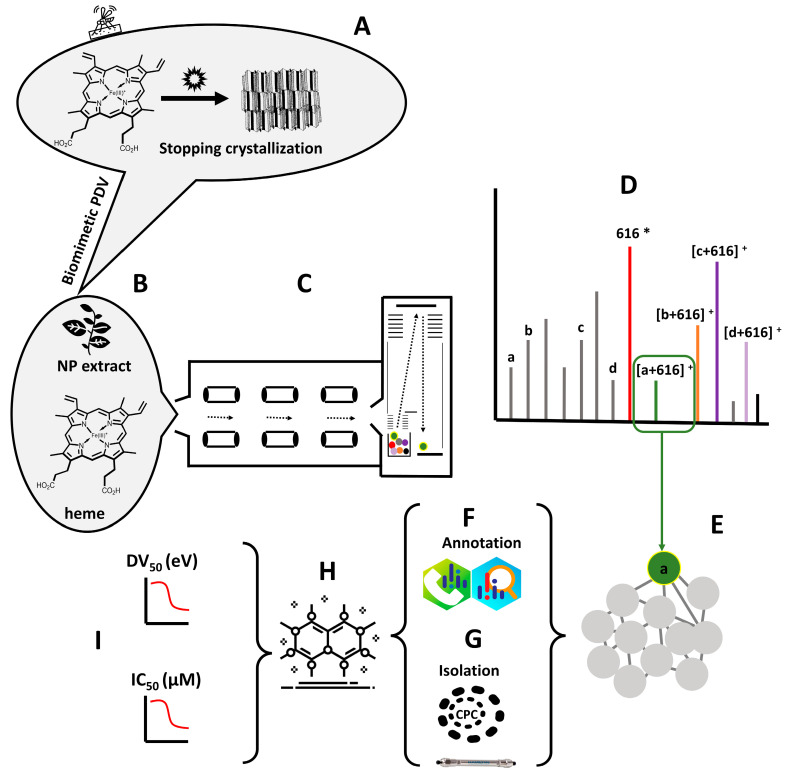
Biodereplication workflow. Schematization of the mechanism of action of antimalarial drugs based on the inhibition of the crystallization pathway of free heme (Fe(III)PPIX) (**A**). Mixture of heme and a natural extract in a biomimetic Plasmodium digestive vacuole (PDV) solution (**B**). Heme-binding assay by mass spectrometry (MS) using a QToF instrument in infusion mode (**C**). Total ion current (TIC) chromatogram displaying masses providing adducts in positive ionization mode (**D**). In **D**, a, b, c, and d = compounds present in the extract. Molecular networking of infusion mode showing adducts of targeted mass, i.e., [a + 616]^+^. Molecular networking is also used to identify targeted mass in LC–MS data from the NP extract (**E**). Comprehensive annotation of targeted mass using MS-Dial and MS-Finder guided by dedicated databases (**F**). Compound isolation of targeted mass using centrifugal partition chromatography (CPC) and preparative HPLC (**G**). Chemical characterization of isolated compounds (**H**). Pharmacological validation of isolated compounds: Heme-binding strength (expressed as DV_50_) and in vitro biological activity (expressed as IC_50_). NP = Natural product; * [heme]^+^ = 616 (**I**).

**Figure 2 molecules-27-07638-f002:**
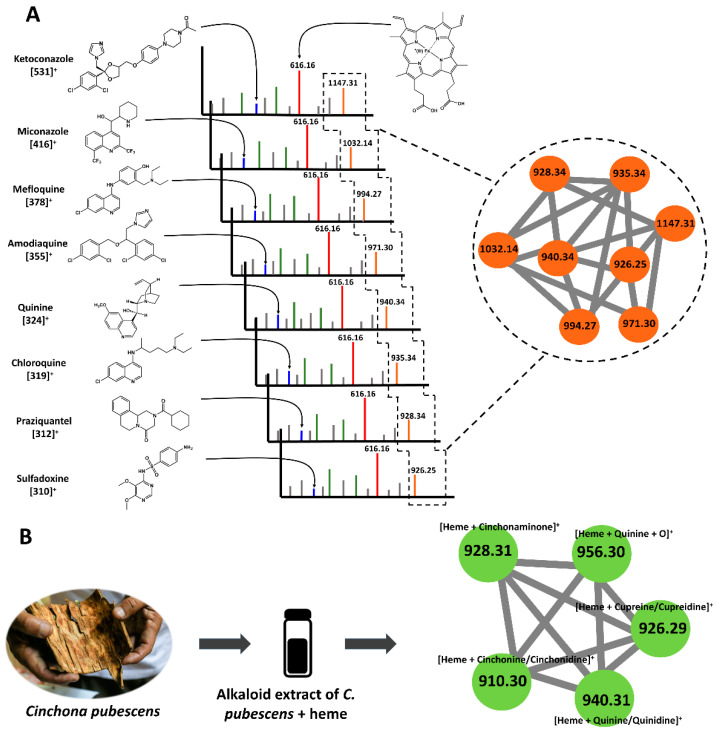
Validation of the biodereplication workflow. (**A**) Visualization by MS and molecular network of heme adducts for a set of antimalarial drugs. Spectrum of each drug–heme mixture displayed the *m*/*z* of drug [M+H]^+^, heme [616]^+^, and drug–heme adduct [drug + heme]^+^. Additionally, MN for all drugs is shown in the same cluster. (**B**) Visualization by MS and molecular network of heme adducts in an alkaloid extract from *C. pubescens*.

**Figure 3 molecules-27-07638-f003:**
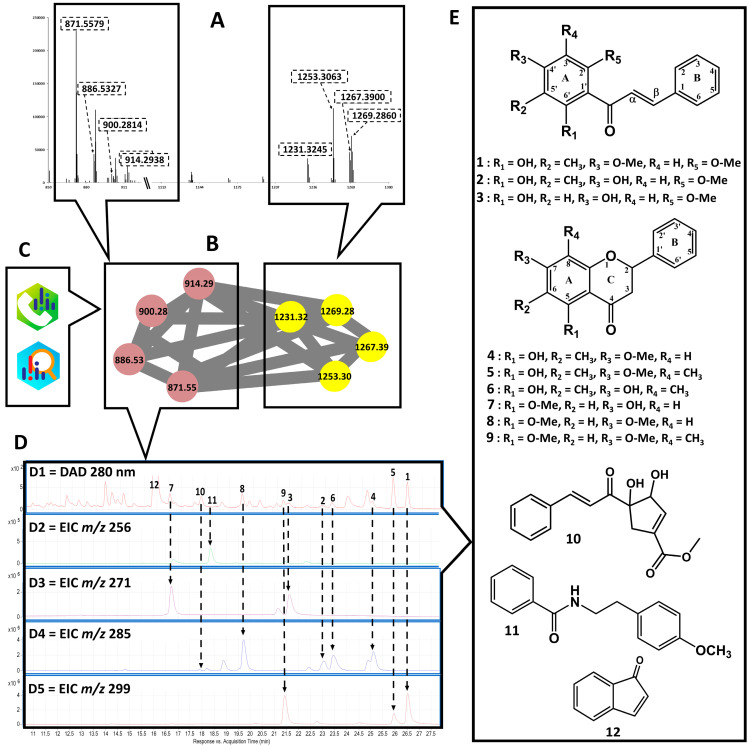
Biodereplication workflow by MS from *P. coruscans* leaves extract. MS spectrum of heme–extract mixture showing the region of adducts (left, dotted square) and the region of heme dimers (right, dotted square) (**A**). Network showing targeted adducts grouped in the same cluster (left square) and heme dimers (right square) (**B**). Annotation of compounds identified using MS-Dial and MS-Finder (**C**). LC-DAD (280 nm) chromatogram of ethanolic extract from *P. coruscans* (**D1**). Extract ion chromatogram (EIC) corresponding to *m*/*z* 256 (**D2**), *m*/*z* 271 (**D3**), *m*/*z* 285 (**D4**), and *m*/*z* 299 (**D5**). Isolated target compounds from *P. coruscans* (**E**).

**Figure 4 molecules-27-07638-f004:**
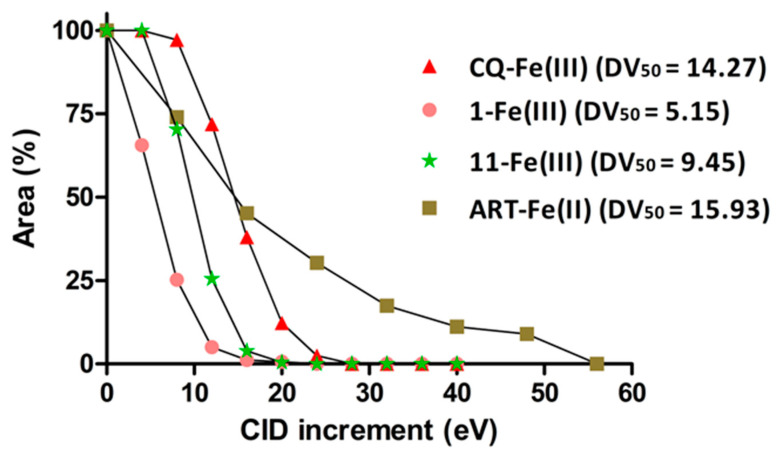
Dissociation curves for the more significant compounds isolated from *P. coruscans* compared with chloroquine (CQ) and artemisinin (ART).

**Table 1 molecules-27-07638-t001:** Details on the annotation of targeted compounds. Using NPClassifire and ClassyFire [48,49], the major classes and subclasses of compounds following their biosynthetic pathway were identified. Cpd = compounds; NA = compounds not annotated; RT = retention time; ** compound* not targeted with the heme-binding assay; ** see Section 3.5.

Cpd	RT	*m*/*z* [M + H]^+^	Formula	Compound Annotated	cLogP	Level Annotation **	Classyfire (Superclass/ Class)
1	26.51	299.123	C_18_H_18_O_4_	aurentiacin	4.47	generic	flavonoid/chalcone
2	23.02	285.110	C_17_H_16_O_4_	2′, 4′-dihydroxy-6′-methoxy-3′-methylchalcone	4.08	generic	flavonoid/chalcone
3	21.61	271.094	C_16_H_14_O_4_	cardamomin	3.62	genus	flavonoid/chalcone
4	25.25	285.111	C_17_H_16_O_4_	strobopinin 7-methyl ether	4.57	generic	flavonoid/flavanones
5	25.98	299.127	C_18_H_18_O_4_	5-hydroxy-7-methoxy-6,8-dimethyl flavanone	5.03	genus	flavonoid/flavanones
6	23.49	285.110	C_17_H_16_O_4_	desmethoxymatteucinol	4.85	generic	flavonoid/flavanones
7	16.75	271.495	C_16_H_14_O_4_	alpinetin	3.71	genus	flavonoid/flavanones
8	19.74	285.109	C_17_H_16_O_4_	pinocembrin	3.40	generic	flavonoid/flavanones
9	21.48	299.127	C_18_H_18_O_4_	dimethyl cryptostrobin	3.86	generic	flavonoid/flavanones
10	17.80	285.111	C_17_H_16_O_4_	NA	2.44	NA	Phenylpropanoi/cinnamic acid and derivative
11	18.23	256.133	C_16_H_17_NO_2_	*N*-Benzoyltyrarnine methyl ether	2.79	generic	alkaloid/alkylamide
12 *	16.01	131.048	C_9_H_6_O	1*H*-inden-1-one	2.21	generic	naphthalenes/naphthoquinone (indanone)

**Table 2 molecules-27-07638-t002:** Antiplasmodial activity, cytotoxicity, and dissociation voltage of isolated compounds. NT= not tested; NA = no adduct; CQ = Chloroquine; ART = Artemisinin. * Adduct not detected. Cpd = compounds. Selectivity index was calculated with CC_50_ on AB943 Primary human fibroblast line cells and IC_50_ for antiplasmodial activity.

Cpd	*m*/*z* [M + H]^+^	Adduct *m*/*z* Expected with heme-Fe (III)	3D7 Sensitive Strain IC_50_ (µM)	AB943 Primary Human Fibroblast CC_50_ (µM)	Selectivity Index CC_50_/IC_50_	DV_50_ (eV)
1	299	915	2.25	68.5	30.4	5.15
2	285	900	51	>100	1.9	5.93
3	271	886	5.5	58	10.5	5.60
4	285	900	60	>100	<1	NA
5	299	915	33.2	23	<1	NA
6	285	900	71	>100	1.4	NA
7	271	886	72	>100	1.4	8.40
8	285	900	78	>100	1.3	8.68
9	299	915	85	>100	1.2	7.68
10	285	900	>100	>100	1	NA
11	256	871	>100	>100	1	9.45
12	131	746 *	>100	>100	1	NA
Crude extract			1.36 ± 0.06 µg/mL	37.5 µg/mL	27.6	NT
CQ	320	935	0.04 ± 3.25	NT	NT	14.27
ART	283	898 (heme-Fe(II)	0.004 ± 0.1 [69]	NT	NT	15.93

## Data Availability

Not applicable.

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
