# Peer review of "Biodereplication of Antiplasmodial Extracts: Application of the Amazonian Medicinal Plant Piper coruscans Kunth"

_molecules, 2022, doi:10.3390/molecules27217638_

Round 1

Reviewer 1 Report

The submitted manuscript describe a highly useful method for fast identification of compounds with antiplasmodial activity. The proposed approach was properly planned and sufficiently documented and it could be used in future high throughput studies focused on search of antimalarial compounds. The amount of provided data and numerous methods supporting the final conclusions is notable, and first of all properly selected. 

Just a few questions / comments:

-        In the introduction the goal of the study could be better highlighted and  more clearly presented. The same applies to the conclusion section of the manuscript

-        In the Discussion part of the manuscript, there are sentences which do not correspond to each other, and should be rephrased - “Given the insolubility of heme in acidic aqueous media…” and “On the other hand, heme is only sparingly soluble in acidic water…”

-        What was the accuracy of m/z measurement? This data could be added in the methodological part of the study.

-        Would it be possible to add a picture of Piper coruscans Kunth (Piperacecae)?

Author Response

  1. In the introduction the goal of the study could be better highlighted and more clearly presented. The same applies to the conclusion section of the manuscript

We amended the introduction with the following paragraph:

“In this work we propose an integrated workflow applicable to natural extracts and dedicated at isolating compounds of biological relevance. We implement several tools to screen an extract for heme-binding compounds, annotate and isolate them in an integrated workflow. This allows an early detection of potential antiplasmodial compounds in crude extracts. Heme binding used as a surrogate of the antiplasmodial activity is monitored by mass spectrometry under biomimetic conditions. Automated annotation of targeted mass through molecular networking and data mining was followed by a mass-guided compound isolation by taking advantage of the versatility and finely tunable selectivity offered by centrifugal partition chromatography. This biodereplication workflow was applied to an extract of Piper coruscans Kunth (Piperaceae)…”

Regarding the conclusion, we sincerely believe we properly described the contribution this work makes on the array of available tools and approaches to discover bioactive natural products. We would prefer to leave the conclusion as it is unless the reviewer has more specific and explicit requests.

  1. In the Discussion part of the manuscript, there are sentences which do not correspond to each other, and should be rephrased - “Given the insolubility of heme in acidic aqueous media…” and “On the other hand, heme is only sparingly soluble in acidic water…”

The sentences cited by the reviewer are not present in the manuscript submitted to MDPI Molecules. We did not find any o inconsistency is the present manuscript regarding solubility of heme.

  1. What was the accuracy of m/z measurement? This data could be added in the methodological part of the study.

In the section 3.5. LC-MS of extract and compounds annotation (line 455 to 456), we had added this paragraph “Resolution of the MS instrument was ca 5,000 at m/z 118 and 10,000 at m/z 922”, about the accuracy of m/z.

  1. Would it be possible to add a picture of Piper coruscans Kunth (Piperacecae)?

A picture of the species has been added in the graphical abstract.

Reviewer 2 Report

This paper is very meaningful to offer a rapid, efficient, and robust biodereplication method to identify potential antimalarial compounds in complex extracts like plant extracts. But I have some issues with the manuscript:

1.Heme binding method have been applied in previous study to evaluate the antimalarial compounds? The mechanism needs further clarification.

2.How to explain the antimalarial activity for compound 3? It is not relying on a heme binding mechanism, but it can be identify by this “biodereplication” method.

3.Three-line table is suggested in the manuscript. Some abbreviation should be checked and consistent.

4.How to understand the meaning of Level annotation of generic or genus in table1.

5. What is meaning of DV50 for antimalarial activity? Why is compound 1 and 3 have relatively lower value but higher activity?

Author Response

1.Heme binding method have been applied in previous study to evaluate the antimalarial compounds? The mechanism needs further clarification.

References 22 to 30 are given in the Introduction section (paragraph from line 77 to 87) to back up the description of methods using heme as a bioassay and the mechanism of action of heme binding. We added the following sentence in this paragraph to explicitly mention the relevance of such a target:

“These methods are based on the assumption that a molecule binding to heme would prevent its crystallization and increase the amount of free heme in the digestive vacuole, hence leading to parasite death”.

2.How to explain the antimalarial activity for compound 3? It is not relying on a heme binding mechanism, but it can be identify by this “biodereplication” method.

During our biodereplication method, we identify a promising adduct with m/z at 886, corresponding to a target m/z 271 (compounds 3 and 7). As the method cannot distinguish isomers, the isolation and the in vitro antiplasmodial activity for these two compounds were performed. In the case of the compound 3, it presented a good activity (IC50 = 5.5 µM) but a low heme-binding ability (DV50 = 5.6) when compared with the positive control chloroquine (DV50 = 14.27). In this work, we focused on a single mechanism of action of antimalarial drugs, the impairment of the free heme detoxification pathway. This is the most documented mechanism of action of quinoline derivatives (chloroquine, amodiaquine, mefloquine, etc.). However, other mechanisms of action can lead to an antiplasmodial effect, as targeting P. falciparum tubulin, folates pathway, mitochondrial functions, ATPase4 carrier, etc. Additionally, one compound may have various antimalarial mechanisms. The following sentence was added in line 324:

“As the isolation of compounds was mass-driven, isomers could not be distinguished, and compounds not binding to heme were isolated as well. An antiplasmodial activity may still be detected for these compounds, albeit not mediated by heme binding”.

3.Three-line table is suggested in the manuscript. Some abbreviation should be checked and consistent.

The tables have been corrected, and the abbreviation have been checked.

  1. How to understand the meaning of Level annotation of generic or genus in table1.

We added a note in the table’s caption to steer the reader to the section 3.5. LC-MS of extract and compounds annotation, which explains the different levels of annotation: “Two pathways were followed for the compound’s annotation by comparing data to 2 dedicated compounds databases: i) compounds identified in the literature for both Piper genus and Piperaceae family (Dictionary of Natural Products version 28.2, CRC press) mined by MS Finder based on exact mass and in silico fragmentation (“genus” annotation level); and ii) generic databases included in MS-FINDER as PlantCyc, ChEBI, NANPDB, COCONUT, and KNApSAcK (“generic” annotation level)”.

  1. What is meaning of DV50 for antimalarial activity? Why is compound 1 and 3 have relatively lower value but higher activity?

DV50 was defined in the manuscript (line 321 and fig 4), and we added a more explicit definition of DV50 in the text. The question on compounds having significant antiplasmodial activity along with a low DV50 of the same nature of question 2 from the same reviewer, thus is addressed by the same added paragraph (line 321):

“Indeed, DV50 is the fragmentor voltage at which half of the adduct is dissociated (see Figure 4). We previously demonstrated that it can be interpreted as proportional to the binding strength between heme and compound. As the isolation of compounds was mass-driven, isomers could not be distinguished, and compounds not binding to heme were isolated as well. An antiplasmodial activity may still be detected for these compounds, albeit not mediated by heme binding”.

Reviewer 3 Report

Antimalarial drug discovery is still an important topic in connection with the resistance problem. This work describes a so called biodereplication method for the quick screening of potential antiplasmodial compounds in Piper coruscans Kunth crude plant extracts based on the heme binding and mass spectrometry assay. This biodereplication method allows a rapid and efficient technique to identify potential antimalarial compounds in complex extracts like plant extracts. It’s a novel method. This reviewer recommend its publication after minor revision.

1)     Abstract:“hit source”?

2)     Page 3, line 113, “Solubility of water..”?

3)     The characterization data in supporting information should be mentioned and/or discussed in the main text, in this connection, the numbering of figures and tables in supporting information should be different from main text, eg Figure S1….Table S1…

Author Response

1)     Abstract:hit source”?

We changed it for “ a source of drug candidates

2)     Page 3, line 113, “Solubility of water..”?

This is correct. Water and octanol are biphasic, yet a fraction of water is solubilized in octanol.

3)     The characterization data in supporting information should be mentioned and/or discussed in the main text, in this connection, the numbering of figures and tables in supporting information should be different from main text, eg Figure S1….Table S1…

We modified the text and added some discussions.

Reviewer 4 Report

Vásquez-Ocmín et al herein presented the article - Biodereplication of antiplasmodial extracts: application to the Amazonian medicinal plant Piper coruscans Kunth. Although the work was intelligently executed and well presented, I have some concerns.

1.  Some grammatical checks should be carried out on the article. For instance, Line 18: the word 'hasten' should replace the word 'fasten'; bounds in line 72 should be bonds and detoxication in line 78 should be detoxification. 

2.  Under section 3.6 (Isolation of targetted compounds), why is the LCMS data not shown (perhaps in the Supporting info)? The Volumes of each fraction are not stated. This is important.

3. Authors should address the plagiarism concern in the attached document so that the originality of the article can be established.

Author Response

  1. Some grammatical checks should be carried out on the article. For instance, Line 18: the word 'hasten' should replace the word 'fasten'; bounds in line 72 should be bonds and detoxication in line 78 should be detoxification.

We applied the corrections as suggested. Nevertheless, it seems that the words “hasten” and “detoxication” are correct.

  1. Under section 3.6 (Isolation of targeted compounds), why is the LCMS data not shown (perhaps in the Supporting info)? The Volumes of each fraction are not stated. This is important.

LCMS data for each compound isolated were added in the Supporting Information (SI10). CPC fraction volume was 0.5 ml, this information (which is given in the experimental section) was added on the figure.

  1. Authors should address the plagiarism concern in the attached document so that the originality of the article can be established.

This report shows a value of 57% of similarity with previous work. However, 36% correspond to the a preprint report of this work (https://chemrxiv.org/engage/chemrxiv/article-details/60c7480bee301c96a0c797f0). Indeed, the preprint publication was a basis for the current article. The other 11% were presented in the PhD thesis manuscript of Pedro Vásquez–Ocmín (first author) which is also the basis of this work. Five other sources account for 5% total, and points to publications of Pedro Vásquez–Ocmín or our group, where important/indispensable information about the methodology by LCMS were described.

Round 2

Reviewer 4 Report

My earlier concerns have been addressed. Considering the response of the authors,  I actually suggested the word hasten.... not fasten....(in the abstract). Good work.

Author Response

We applied the correction as suggested in the abstract.